# Trends in and Risk Factors for Drug Resistance in *Mycobacterium tuberculosis* in HIV-Infected Patients

**DOI:** 10.3390/v16040627

**Published:** 2024-04-18

**Authors:** Xiaoqin Le, Xueqin Qian, Li Liu, Jianjun Sun, Wei Song, Tangkai Qi, Zhenyan Wang, Yang Tang, Shuibao Xu, Junyang Yang, Jiangrong Wang, Jun Chen, Renfang Zhang, Zhaoqin Zhu, Yinzhong Shen

**Affiliations:** 1Department of Infection and Immunity, Shanghai Public Health Clinical Center, Fudan University, Shanghai 201508, China; 2Department of Clinical Laboratory, Shanghai Public Health Clinical Center, Fudan University, Shanghai 201508, China

**Keywords:** HIV, *Mycobacterium tuberculosis*, quinolone, rifampicin

## Abstract

Trends in and risk factors for drug resistance in *Mycobacterium tuberculosis* (*M. tuberculosis*) in human immunodeficiency virus (HIV)-infected patients with active tuberculosis were analyzed. The clinical data of *M. tuberculosis* and HIV-coinfected patients treated at the Shanghai Public Health Clinical Center between 2010 and 2022 were collected. The diagnosis of tuberculosis was confirmed by solid or liquid culture. The phenotypic drug susceptibility test was carried out via the proportional method, and the resistance to first-line and second-line drugs was analyzed. Logistic regression analysis was performed to identify associated risk factors for drug resistance in *M. tuberculosis*. Of the 304 patients with a *M. tuberculosis*-positive culture and first-line drug susceptibility test results, 114 (37.5%) were resistant to at least one first-line anti-tuberculosis drug. Of the 93 patients with first-line and second-line drug susceptibility test results, 40 (43%) were resistant to at least one anti-tuberculosis drug, and 20 (21.5%), 27 (29.0%), 19 (20.4%), 16 (17.2%), and 14 (15.1%) were resistant to rifampicin, streptomycin, ofloxacin, levofloxacin, and moxifloxacin, respectively; 17 patients (18.3%) had multidrug-resistant tuberculosis (MDR-TB). Between 2010 and 2021, the rate of resistance to streptomycin and rifampicin ranged from 14.3% to 40.0% and from 8.0% to 26.3%, respectively, showing an increasing trend year by year. From 2016 to 2021, the rate of resistance to quinolones fluctuated between 7.7% and 27.8%, exhibiting an overall upward trend. Logistic regression analysis showed that being aged <60 years old was a risk factor for streptomycin resistance, mono-drug resistance, and any-drug resistance (RR 4.139, *p* = 0.023; RR 7.734, *p* = 0.047; RR 3.733, *p* = 0.009). Retreatment tuberculosis was a risk factor for resistance to rifampicin, ofloxacin, of levofloxacin (RR 2.984, *p* = 0.047; RR 4.517, *p* = 0.038; RR 6.277, *p* = 0.014). The drug resistance rates of *M. tuberculosis* to rifampicin and to quinolones in HIV/AIDS patients were high and have been increasing year by year. Age and a history of previous anti-tuberculosis treatment were the main factors associated with the development of drug resistance in HIV/AIDS patients with tuberculosis.

## 1. Introduction

Human immunodeficiency virus (HIV)-infected individuals, due to severe impairment of the immune function, are prone to various opportunistic infections. Tuberculosis is a common opportunistic infection and is the major cause of death among HIV/AIDS patients [1]. In 2022, tuberculosis caused nearly 167,000 deaths in HIV-infected patients globally, constituting over 25% HIV-related mortalities [2,3]. Although the global incidence and mortality rate of tuberculosis among HIV/AIDS patients have steadily declined between 2010 and 2022, the annual incidence of rifampicin-resistant and multidrug-resistant tuberculosis (RR/MDR-TB) has remained persistently high, especially between 2020 and 2022 [2]. Globally, there were 410,000 incident cases of RR/MDR-TB in 2022, with China contributing 30,000 cases during this period, representing nearly 7.3% [2,4]. According to the global tuberculosis report of 2023 by WHO, the treatment success rate was 88% for drug-susceptible tuberculosis in 2021, 79% among HIV/AIDS patients in 2021, and 63% for RR/MDR-TB in 2020, indicating that the anti-tuberculosis treatment’s success rate was lower in patients with RR/MDR-TB and those co-infected with HIV [2]. Given the prevailing high prevalence of drug-resistant tuberculosis, coupled with its low treatment success rate, there is an urgent need to elucidate the resistance profiles of tuberculosis in HIV/AIDS patients in China, so as to provide evidence for taking effective prevention and control measures. Although several previous domestic studies have addressed this issue, the findings were inconsistent, and longitudinal data on annual drug resistance trends were scarce. To address this gap and investigate the drug resistance patterns of tuberculosis in recent years, we conducted a study involving HIV-positive patients treated at the Shanghai Public Health Clinical Center between 2010 and 2022. We collected clinical data and drug susceptibility test results. The study’s primary objectives were to analyze drug resistance characteristics, annual trends, and the associated risk factors.

## 2. Materials and Methods

### 2.1. Study Population

HIV and *Mycobacterium tuberculosis* (*M. tuberculosis*) co-infected patients (aged ≥18 years old) treated at the Shanghai Public Health Clinical Center affiliated with Fudan University between 2010 and 2022 were included in this study if the mycobacterial culture isolate was identified as *M. tuberculosis* and the results of anti-tuberculosis drug susceptibility tests were available. Clinical data, including gender, age, CD4^+^ T lymphocyte count, and status of anti-tuberculosis treatment (initial treatment or retreatment), were collected for analysis. Informed consent was obtained from all the patients, and the study was approved by the Ethics Committee of Shanghai Public Health Clinical Center (protocol code: 2021-S051-03, approved on 27 January 2022).

HIV infection/AIDS was diagnosed according to *Chinese Guidelines for Diagnosis and Treatment of HIV/AIDS* (2021 edition) [5]. Diagnosis of tuberculosis and drug resistance classification were based on WHO guidelines and the Chinese Classification of Tuberculosis (WS 196—2017) [2,6,7]. The criteria for initial treatment and retreatment tuberculosis were outlined as follows. Patients were categorized as having initial treatment of tuberculosis if they fulfilled at least one of the following criteria: (a) patients who were newly diagnosed with tuberculosis and had not received any prior treatment with anti-tuberculosis drugs; (b) patients who had initiated a standard course of anti-tuberculosis treatment but had not completed it; and (c) patients who had undergone an irregular regimen of anti-tuberculosis treatment for a period shorter than 1 month. This could involve irregular doses, skipped doses, or the use of non-standardized regimens. Patients were classified as having retreatment tuberculosis if they meet either of the following conditions: (a) patients who had a history of receiving anti-tuberculosis drugs for at least 1 month, but their treatment was irrational or irregular; and (b) patients who experienced failure of the initial treatment or a relapse of tuberculosis.

### 2.2. Methods

#### 2.2.1. Specimen Collection and Culture

Specimens, including sputum, pleural effusion, ascites fluid, urine, feces, cerebrospinal fluid, peripheral blood, lymphoid tissue, and surgical excisions, were collected from the patients. If the strains isolated from various specimens of the same patient showed identical bacterial typing and drug susceptibility characteristics, we counted them as one case for statistical purposes. Non-blood specimens were cultured using Roche medium (Besso Corporation, Zhuhai, Guangdong, China) and/or BACTEC MGIT960 liquid medium (BD Company, Bergen, NJ, USA). Blood specimens were cultured using the BACTEC 9120 blood culture system (BD Company, Bergen, NJ, USA); if the culture was positive, the isolated strains were transferred to Roche medium for further confirmation. All culture-positive results were confirmed by acid-fast staining. Finally, strain identification was performed using the MPB64 antigen immunocolloidal gold detection kit (Hangzhou Genesis Corporation, Hangzhou, Zhejiang, China). The strain type was considered as *M. tuberculosis* only when both the specimen culture and MPB64 antigen test were positive.

#### 2.2.2. Drug Susceptibility Test

The proportion method was used for phenotypic susceptibility testing, including 4 first-line and 8 s-line drugs, based on the Laboratory Testing Procedures for Tuberculosis Diagnostics [8]. During the period from 2010 to 2015, drug susceptibility testing in our hospital only included 4 first-line anti-tuberculosis drugs (isoniazid, rifampicin, ethambutol, and streptomycin). The final concentrations of each drug in the culture medium were as follows: isoniazid, 0.2 μg/mL; rifampicin, 40 μg/mL; ethambutol, 2 μg/mL; streptomycin, 4 μg/mL; kanamycin, 30 μg/mL; capreomycin, 40 μg/mL; para-aminosalicylic acid, 1 μg/mL; protionamide, 40 μg/mL; amikacin, 30 μg/mL; ofloxacin, 4 μg/mL; levofloxacin, 2 μg/mL; moxifloxacin, 1 μg/mL. If there is a discrepancy in the drug sensitivity results between the solid and liquid methods, preference was given to the results from the solid method. The *M. tuberculosis* standard strain H37Rv (ATCC 27294) was used as the positive control in every batch of bacterial strain identification and drug sensitivity tests.

### 2.3. Definition of Drug-Resistant Tuberculosis

Drug-resistant tuberculosis (DR-TB) refers to tuberculosis that has resistance to at least one anti-tuberculosis drug.Mono-resistant tuberculosis (MR-TB) refers to tuberculosis that has resistance to only one anti-tuberculosis drug.Isoniazid-resistant tuberculosis (Hr-TB) refers to tuberculosis that has resistance to isoniazid but not concurrent resistance to rifampicin.Rifampicin-resistant tuberculosis (RR-TB) refers to tuberculosis that has resistance to rifampicin regardless of its sensitivity or resistance to other anti-tuberculosis drugs.Poly-resistant tuberculosis (PR-TB) refers to tuberculosis that has resistance to more than one kind of anti-tuberculosis drug, but not concurrent resistance to isoniazid and rifampicin.Multidrug-resistant tuberculosis (MDR-TB) refers to tuberculosis that has simultaneous resistance to isoniazid and rifampicin, regardless of its resistance to other anti-tuberculosis drugs.Pre-extensive drug-resistant tuberculosis (Pre-XDR-TB) refers to MDR-TB that has resistance to any quinolone.Extensive drug resistant tuberculosis (XDR-TB) refers to MDR-TB that has resistance to any quinolone and at least one of the three injectable drugs including capreomycin, kanamycin, and amikacin [6]. The traditional definition of XDR-TB was still used in this study because linezolid and bedaquiline were unavailable clinically and phenotypic susceptibility testing was not carried out in our hospital.

### 2.4. Statistical Analysis

SPSS 26.0 software was used for data analysis. Continuous variables were expressed as the median (inter-quartile range) or the mean ± standard deviations, and categorical variables were expressed as proportions and frequencies. Comparisons between groups were conducted using Student’s *t*-test or Wilcoxon’s rank-sum test for continuous variables, and the χ^2^ test or Fisher’s exact test for categorical variables. Spearman’s correlation analysis was performed. All variables with a *p*-value of <0.2 in Spearman’s correlation analysis as well as the factors that could have a clinically significant impact on drug resistance patterns were entered in the binary logistic regression to determine the independent risk factors associated with drug-resistant tuberculosis. A *p*-value of <0.05 was considered statistically significant.

## 3. Results

### 3.1. The Overall Profiles of Drug Resistance

Of the 675 patients with culture-positive tuberculosis, 304 received first-line anti-tuberculosis drug susceptibility testing and 93 also underwent second-line drug susceptibility testing.

Of the 304 patients with first-line drug susceptibility results, 94.1% (286/304) had initial treatment tuberculosis and 37.5% (114/304) were resistant to at least one first-line anti-tuberculosis drug. Of the 114 patients with drug-resistant tuberculosis, 89.5% (102/114) were male and 95.6% (109/114) were <60 years old, with an average age of 39.8 ± 11.7 years. The resistance rates of rifampicin and streptomycin were 15.1% (46/304) and 26.6% (81/304), with the MDR-TB rate being 13.2% (40/304) (Table 1). Among the 286 initial treatment and 18 retreatment patients, the respective rates of any-drug resistance, rifampicin resistance, streptomycin resistance, and MDR-TB were as follows: 37.4% (107/286) vs. 38.9% (7/18), 14.0% (40/286) vs. 33.3% (6/18), 26.9% (77/286) vs. 22.2% (4/18), and 12.6% (36/286) vs. 22.2% (4/18), respectively (Table 2).

Of the 93 patients with first-line and second-line drug susceptibility results, 89.2% (83/93) had initial treatment of tuberculosis and 43% (40/93) were resistant to at least one anti-tuberculosis drug. Of the 40 drug-resistant patients, 92.5% (37/40) were male and 92.5% (37/40) were <60 years old, with an average age of 40.6 ± 11.6 years. The resistance rates of rifampicin, streptomycin, ofloxacin, levofloxacin, and moxifloxacin were 21.5% (20/93), 29.0% (27/93), 20.4% (19/93), 17.2% (16/93), and 15.1% (14/93), respectively. The rates of MDR-TB, pre-XDR-TB, and XDR-TB were 18.3% (17/93), 15.1% (14/93), and 5.4% (5/93), respectively (Table 1). Among the 83 initial-treatment patients and 10 retreatment patients, the resistance rates of rifampicin, streptomycin, ofloxacin, levofloxacin, and moxifloxacin were 18.1% (15/83) vs. 50% (5/10), 27.7% (23/83) vs. 40% (4/10), 16.9% (14/83) vs. 50% (5/10), 13.3% (11/83) vs. 50% (5/10), and 13.3% (11/83) vs. 30% (3/10), respectively; the rate of MDR-TB was 15.7% (13/83) in the initial-treatment patients and 40% (4/10) in the retreatment patients (Table 3).

### 3.2. The Trends in Drug Resistance in M. tuberculosis to the First-Line and Second-Line Anti-Tuberculosis Drugs from 2010 to 2021

Between 2010 and 2021 (only four cases in 2022 underwent susceptibility testing and thus were excluded from the analysis), the resistance rates of streptomycin and rifampicin ranged from 14.3% to 40.0% and from 8.0% to 26.3%, respectively, demonstrating annual increases throughout the period (Figure 1). From 2016 to 2021, the rate of resistance to quinolones fluctuated between 7.7% and 27.8%, exhibiting an overall upward trend (Figure 2).

### 3.3. Comparison of Drug Resistance Rates between Different Groups and Risk Factor Analysis in the Group of 304 Cases

The 304 patients with four first-line drug susceptibility results were divided into two groups on the basis of gender, age, CD4 cell count, and history of anti-tuberculosis treatment to compare the differences in the resistance rates. The results showed that the rates of streptomycin resistance, mono-drug resistance, and any-drug resistance were higher among cases aged <60 years old compared with the control group (28.8% (78/271) vs. 9.1% (3/33), 19.6% (53/271) vs. 3.0% (1/33), and 40.2% (109/271) vs. 15.2% (5/33)), with all *p*-values being <0.05. The rifampicin resistance rate was higher in retreatment patients than in the initial-treatment patients (33.3% (6/18) vs. 14% (40/286)), albeit with a marginally non-significant difference (*p* = 0.06) (Table 2). The binary logistic regression showed that being aged <60 years old was the risk factor for streptomycin resistance, mono-drug resistance, and any-drug resistance (RR 4.139, *p* = 0.023; RR 7.734, *p* = 0.047; RR 3.733, *p* = 0.009); retreatment tuberculosis was identified as a risk factor for rifampicin resistance (RR 2.984, *p* = 0.047) (Figure 3, Figure 4, Figure 5 and Figure 6). 

### 3.4. Comparison of Drug Resistance Rates between Different Groups and Risk Factor Analysis in the Group of 93 Cases 

The 93 patients with four first-line and eight second-line drug susceptibility results were divided into two groups on the basis of gender, age, CD4 cell count, and status of anti-tuberculosis treatment to compare the difference in the resistance rates. The results showed that the any-drug resistance rate was higher in cases aged <60 years old than in the control group (46.8% vs. 21.4%), albeit with a marginally non-significant difference (*p* = 0.077). The resistance rates of kanamycin, protionamide, ofloxacin, and levofloxacin were higher in the retreatment group than in the control group (30% vs. 1.2%, 30% vs. 3.6%, 50% vs. 16.9%, and 50% vs. 13.3%) (all *p* < 0.05). The resistance rates of rifampicin and amikacin were higher in retreatment patients than in the initial-treatment patients (50% vs. 18.1% and 20% vs. 2.4%), with a marginally non-significant difference (both *p*-value = 0.056) (Table 3). The binary logistic regression showed that retreatment tuberculosis was the risk factor for both ofloxacin and levofloxacin resistance (RR 4.517, *p* = 0.038; RR 6.277, *p* = 0.014) (Figure 7 and Figure 8).

## 4. Discussion

The latest research report from WHO indicates that China is a high-burden country for tuberculosis, HIV/*M. tuberculosis* co-infection, and RR/MDR-TB, with the largest number of MDR-TB cases worldwide [2]. Drug resistance in tuberculosis is serious both in the general population and among HIV-infected individuals. Our research revealed that the rate of resistance to at least one anti-tuberculosis drug in HIV/AIDS patients ranged from 37.5% to 43%, which was relatively close to the rate observed in HIV/*M. tuberculosis* co-infected patients in Xinjiang [9] but was higher than the reported rates in other provinces of China, such as Guangxi [10], Sichuan [11], and Henan [12]. The high drug resistance rate was partly associated with the characteristics of the cases enrolled in this study, which included tuberculosis patients who were referred from other provinces and those who underwent surgery. In both situations, a relatively high proportion of patients were infected with drug-resistant tuberculosis. In addition, among the initial-treatment and retreatment tuberculosis cases, the rates of MDR-TB ranged from 12.6% to 15.7% and from 22.2% to 40%, respectively, both of which were higher than the corresponding rates observed in HIV-negative patients in China (5.7% for initial-treatment tuberculosis and 25.6% for retreatment tuberculosis [13]). Several meta-analyses in recent years have demonstrated that HIV infection itself is an important risk factor for the development of MDR-TB [14,15,16], meaning that co-infection with HIV increases the risk of acquiring MDR-TB. The reasons for this may include the increased susceptibility to drug-resistant strains among HIV-positive individuals, the poor absorption of anti-tuberculosis drugs, and poor treatment adherence [17].

In our study, the resistance to rifampicin ranged from 15.1% to 21.5%, while the prevalence of MDR-TB varied between 13.2% and 18.3%. The close proximity of these two ranges suggests that rifampicin resistance, to some extent, reflects the overall situation regarding MDR-TB. In the current analysis, the rank order of drug resistance rates was found to be rifampicin > isoniazid, which differed from the more commonly reported pattern in the general population and other regions, in which the isoniazid resistance rate typically exceeds that of rifampicin resistance. The trend over the years indicates that both the rifampicin resistance rate and the MDR-TB rate have generally been on the rise, suggesting a severe situation concerning the occurrence of RR/MDR-TB among HIV-infected patients in recent years. This finding aligns with data published by WHO and reports from certain provinces in China [2,12]. This poses new challenges to the diagnosis and treatment of tuberculosis, emphasizing the crucial importance of performing drug resistance testing for newly diagnosed tuberculosis. Furthermore, it underscores the necessity of introducing novel drugs against RR/MDR-TB such as linezolid and bedaquiline into clinical practice. The risk factor analysis revealed that the risk of rifampicin resistance increased by 2.984-fold in patients with retreatment tuberculosis, indicating that a history of previous anti-tuberculosis treatment increases the risk of rifampicin-resistant tuberculosis. In the areas with a high prevalence of tuberculosis, immunocompromised individuals with HIV infections are more likely to harbor drug-resistant strains and experience mixed infections with various strains [14,17,18]. Prolonged irrational or irregular anti-tuberculosis treatment facilitates the selection of drug-resistant strains, thereby increasing the risk of RR/MDR-TB among patients with retreatment tuberculosis. Therefore, clinicians should standardize the anti-tuberculosis treatment for HIV-infected individuals with tuberculosis, enhance adherence to the treatment guidelines [1,7], and strengthen patient education on compliance with the treatment in order to improve the success rate of the initial treatment and reduce the emergence of drug resistance. In addition, HIV/*M. tuberculosis* co-infected patients receive multiple agents to treat concomitant diseases, which can impose a heavy burden on the gastrointestinal tract. Some studies suggest that due to chronic diarrhea during advanced HIV infections, the absorption of rifampicin may be compromised. Moreover, significant challenges in managing tuberculosis among HIV-infected patients include the frequent occurrence of adverse events associated with anti-tuberculosis drugs, which often lead to treatment interruptions or dose reductions. Notably, drug-drug interactions between rifamycins and concomitant medications are also commonplace, further complicating therapeutic management. In all the situations above, effective drug concentrations of rifampicin may reduce and lead to acquired resistance. Therefore, therapeutic drug monitoring is recommended under suitable conditions to ensure therapeutic levels of rifampicin. Additionally, the role of the social determinants of health, including low education, low income, and alcohol abuse, has been demonstrated to be associated with increased risks of treatment failure and the development of MDR-TB [19]; however, the extent to which these same determinants exert analogous effects on the outcomes of tuberculosis treatment and drug resistance profiles in HIV-infected patients warrants further investigation. The highest rate of resistance among first-line anti-tuberculosis drugs was observed for streptomycin, with a resistance rate of over 22% reported both in newly treated and retreated patients. The risk factor analysis indicated that patients under the age of 60 years were more prone to developing streptomycin resistance, a finding potentially linked to the relatively younger age distribution of the participants in this study. Whether this elevated susceptibility represents the inherent natural resistance of *M. tuberculosis* to streptomycin necessitates further investigation. Given the high prevalence of streptomycin resistance, its use as a primary component in first-line anti-tuberculosis regimens is currently discouraged in clinical practice.

Among the second-line drugs, quinolones had the highest resistance rates, all exceeding 15%, and the resistance rates in retreated patients were significantly higher than those in newly treated patients. The resistance to quinolones, including ofloxacin, levofloxacin, and moxifloxacin, has increased between the years 2017 and 2019 before the COVID-19 pandemic, which has been confirmed in other studies [9,11]. The serious situation of increasing quinolone resistance might be partially attributed to the frequent misdiagnosis of tuberculosis as common pneumonia, which subsequently leads to the unrestricted empirical use of quinolones. Thus, it is urgent to strengthen the management of quinolones and reduce misuse. Some scholars have pointed out that quinolones should not be used as first-line drugs for common pneumonia in areas with a high prevalence of tuberculosis [20]. Furthermore, the clinical capacity for diagnosing tuberculosis should be improved to ensure early and accurate diagnosis, avoiding unnecessary exposure to quinolones as a mono-therapy, thereby reducing the risk of inducible resistance. The risk factor analysis showed that retreated patients exhibited a 4.517-fold and 6.277-fold increased risk of resistance to ofloxacin and levofloxacin, respectively. This indicates that a history of previous anti-tuberculosis treatment not only predisposes patients to the development of MDR-TB, but also exacerbates the emergence of pre-XDR-TB and even XDR-TB. Therefore, standardizing the use of quinolones, improving the capacity for diagnosing tuberculosis, and enhancing the success rate of the initial treatment are crucial measures for effectively curtailing the occurrence of MDR-TB and XDR-TB.

Our study has several limitations. Firstly, as a single-center investigation, the high resistance rate observed might only be representative of the patients treated at this particular center and may not be generalizable to other settings. Secondly, the relatively small number of cases, particularly when analyzing resistance trends over the years with limited data, hindered a robust assessment of the patterns of changes in resistance; a larger sample size would be required in future studies to provide more definitive insights into such trends. Thirdly, due to the retrospective nature of this study, valuable information on prior antibiotic use within the 6 months preceding the anti-tuberculosis therapy, which could have been included in the risk factor analysis, was unavailable. Nonetheless, similar findings regarding the impact of prior antibiotic exposure have been substantiated by another study [20]. Lastly, social determinants such as socioeconomic status, living conditions, substance misuse behaviors (e.g., alcoholism or drug abuse), and educational achievement, which have been regarded as potential risk factors for drug resistance, were not explored in our study but are indeed deserving of further investigation.

## 5. Conclusions

This study analyzed the trends in and risk factors for the drug resistance of *M. tuberculosis* in HIV/AIDS patients from 2010 to 2022. The results revealed that the rates of resistance to rifampicin and quinolones were high and showed a consistently increasing trend among HIV/AIDS patients with tuberculosis. Age and a history of previous anti-tuberculosis treatment were identified as the major risk factors influencing the development of resistance in *M. tuberculosis*. These findings indicate that the situation regarding drug-resistant tuberculosis in HIV-infected patients is becoming increasingly severe. To address this issue, it is imperative to enhance diagnostic capabilities for tuberculosis, strengthen the standardized use of anti-tuberculosis drugs, and minimize exposure to these drugs in an unregulated manner. Such measures aim to reduce the incidence of drug-resistant tuberculosis and effectively control its spread, thereby laying the groundwork for ultimately achieving the goal of eradicating tuberculosis.

## Figures and Tables

**Figure 1 viruses-16-00627-f001:**
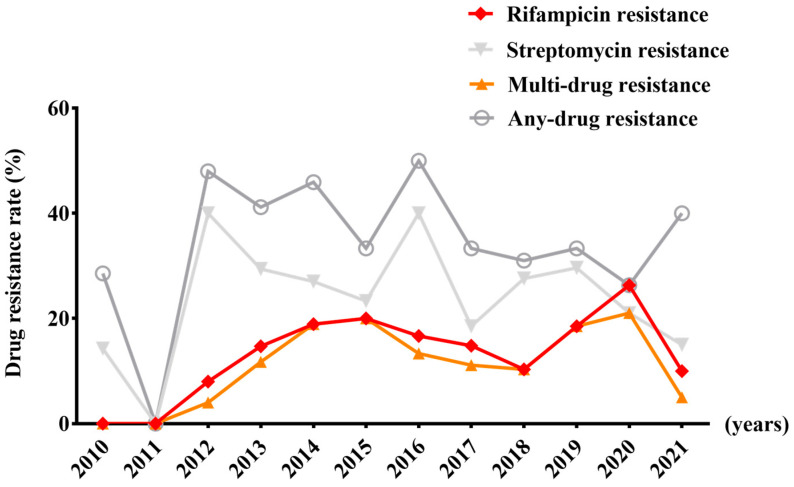
The trends in the rates of first-line drug resistance from 2010 to 2021.

**Figure 2 viruses-16-00627-f002:**
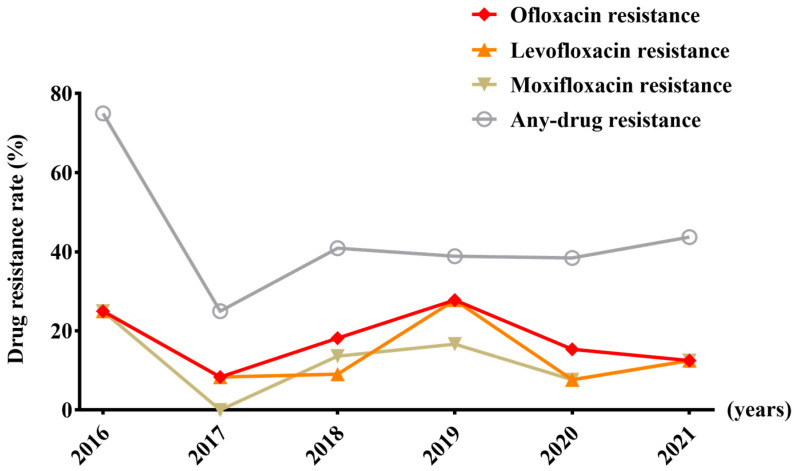
The trends in the rates of second-line drug resistance from 2016 to 2021.

**Figure 3 viruses-16-00627-f003:**
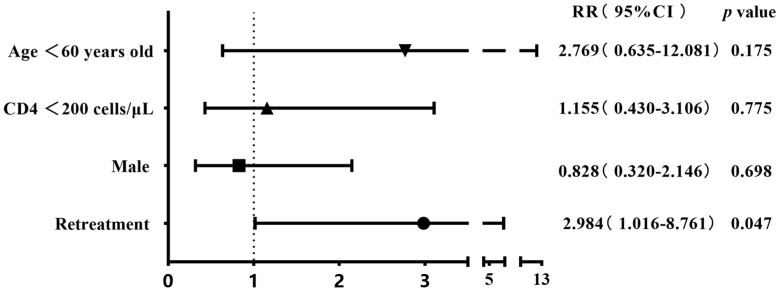
Risk factors associated with rifampicin resistance in the binary logistic regression analysis (n = 304).

**Figure 4 viruses-16-00627-f004:**
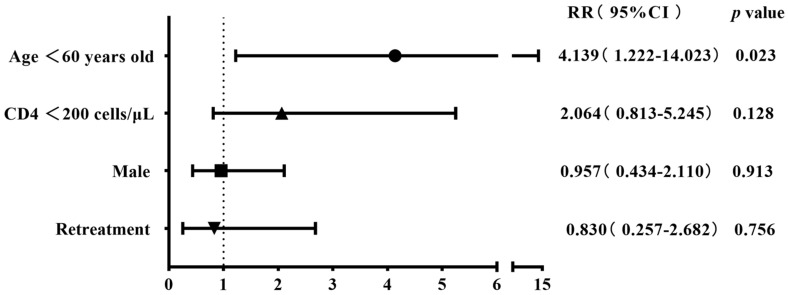
Risk factors associated with streptomycin resistance in the binary logistic regression analysis (n = 304).

**Figure 5 viruses-16-00627-f005:**
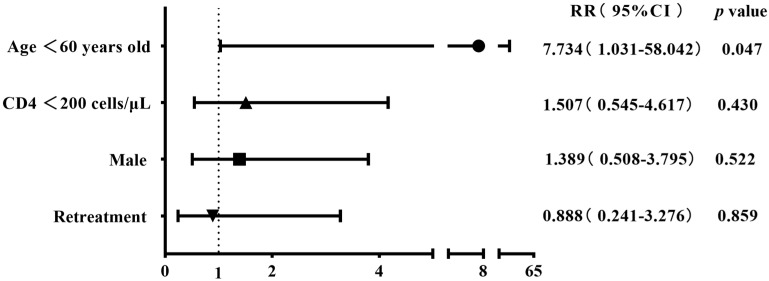
Risk factors associated with mono-drug resistance in the binary logistic regression analysis (n = 304).

**Figure 6 viruses-16-00627-f006:**
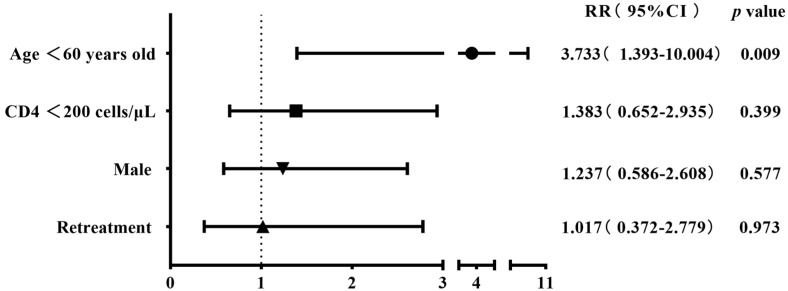
Risk factors associated with any-drug resistance in the binary logistic regression analysis (n = 304).

**Figure 7 viruses-16-00627-f007:**
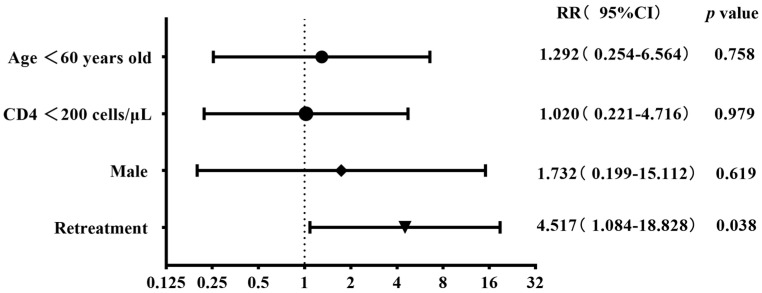
Risk factors associated with ofloxacin resistance in the binary logistic regression analysis (n = 93).

**Figure 8 viruses-16-00627-f008:**
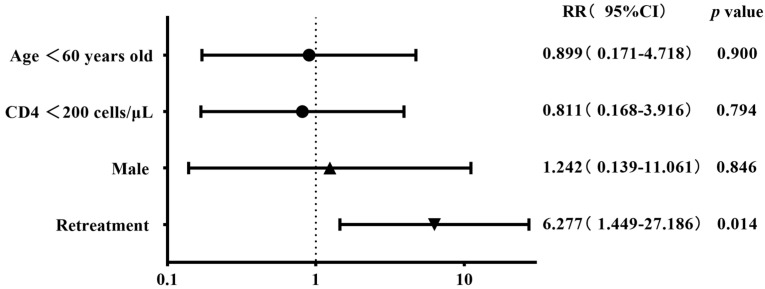
Risk factors associated with levofloxacin resistance in the binary logistic regression analysis (n = 93).

**Table 1 viruses-16-00627-t001:** The profiles of the resistance of *M. tuberculosis* to first-line and second-line anti-tuberculosis drugs.

	Patients with First-Line Drug Susceptibility Results (n = 304)	Patients with First-Line and Second-Line Drug Susceptibility Results (n = 93)
Age ≥ 60 years old, n (%)	33 (10.9)	14 (15.1)
Male, n (%)	267 (87.8)	84 (90.3)
CD4 T cell counts ≥ 200 cells/μL, n (%)	38 (12.5)	12 (12.9)
Initial treatment, n (%)	286 (94.1)	83 (89.2)
Isoniazid resistance, n (%)	35 (11.5)	9 (9.7)
Rifampicin resistance, n (%)	46 (15.1)	20 (21.5)
Ethambutol resistance, n (%)	38 (12.5)	10 (10.8)
Streptomycin resistance, n (%)	81 (26.6)	27 (29)
Kanamycin resistance, n (%)	/	4 (4.3)
Capreomycin resistance, n (%)	/	5 (5.4)
Para-aminosalicylic acid resistance, n (%)	/	7 (7.5)
Protionamide resistance, n (%)	/	6 (6.5)
Amikacin resistance, n (%)	/	4 (4.3)
Ofloxacin resistance, n (%)	/	19 (20.4)
Levofloxacin resistance, n (%)	/	16 (17.2)
Moxifloxacin resistance, n (%)	/	14 (15.1)
XDR-TB, n (%)	/	5 (5.4)
Pre-XDR-TB, n (%)	/	14 (15.1)
MDR-TB, n (%)	40 (13.2)	17 (18.3)
PR-TB, n (%)	20 (6.6)	11 (11.8)
MR-TB, n (%)	54 (17.8)	12 (12.9)
DR-TB, n (%)	114 (37.5)	40 (43.0)

XDR-TB, extensive drug-resistant tuberculosis; pre-XDR-TB, pre-extensive drug-resistant tuberculosis; MDR-TB, multidrug-resistant tuberculosis; PR-TB, poly-resistant tuberculosis; MR-TB, mono-resistant tuberculosis; DR-TB, drug-resistant tuberculosis.

**Table 2 viruses-16-00627-t002:** Comparison of the first-line drug resistance rates between different groups (n = 304).

	Gender				Age				CD4 T Cell Count (Cells/μL)				Treatment History			
	Male	Female	χ^2^	*p* Value	≥60 years old	<60 years old	χ^2^	*p*-Value	≥200	<200	χ^2^	*p*-Value	Initial Treatment	Retreatment	χ^2^	*p*-Value
	(n = 267)	(n = 37)			(n = 33)	(n = 271)			(n = 38)	(n = 266)			(n = 286)	(n = 18)		
Isoniazid resistance rate	12.40%	5.40%	0.94	0.333	9.10%	11.80%	0.03	0.863	13.20%	11.30%	0.005	0.946	11.90%	5.60%	0.2	0.663
Rifampicin resistance rate	15.00%	16.20%	0.04	0.844	6.10%	16.20%	1.646	0.2	15.80%	15.00%	0.015	0.904	14.00%	33.30%	3.5	0.06
Streptomycin resistance rate	26.60%	27.00%	0	0.955	9.10%	28.80%	5.836	0.016	15.80%	28.20%	2.618	0.106	26.90%	22.20%	0	0.871
Ethambutol resistance rate	11.60%	18.90%	0.99	0.32	9.10%	12.90%	0.121	0.728	13.20%	12.40%	0	1	12.60%	11.10%	0	1
Prevalence of MDR-TB	12.70%	16.20%	0.11	0.743	6.10%	14.00%	1.009	0.315	10.50%	13.50%	0.263	0.608	12.60%	22.20%	0.7	0.416
Prevalence of PR-TB	7.10%	2.70%	0.44	0.509	6.10%	6.60%	0	1	7.90%	6.40%	0	1	7.00%	0.00%	0.5	0.502
Prevalence of MR-TB	18.40%	13.50%	0.52	0.47	3.00%	19.60%	5.501	0.019	13.20%	18.40%	0.631	0.427	17.80%	16.70%	0	1
Prevalence of DR-TB	38.20%	32.40%	0.46	0.497	15.20%	40.20%	7.889	0.005	31.60%	38.30%	0.65	0.42	37.40%	38.90%	0.2	0.9

MDR-TB, multidrug-resistant tuberculosis; PR-TB, poly-resistant tuberculosis; MR-TB, mono-resistant tuberculosis; DR-TB, drug-resistant tuberculosis.

**Table 3 viruses-16-00627-t003:** Comparison of the first-line and second-line anti-tuberculosis drug resistance rates between different groups (n = 93).

	Gender				Age				CD4 T Cell Count (Cells/μL)				Condition of Treatment			
	Male (n = 84)	Female (n = 9)	χ^2^	*p*-Value	≥60 Years Old (n = 14)	<60 Years Old (n = 79)	χ^2^	*p*-Value	≥200 (n = 12)	<200 (n = 81)	χ^2^	*p*-Value	Initial treatment (n = 83)	Retreatment (n = 10)	χ^2^	*p*-Value
Isoniazid resistance rate	9.5%	11.1%	-	1	7.1%	10.1%	0	1	8.3%	9.9%	0	1	9.6%	10%	-	1
Rifampicin resistance rate	22.6%	11.1%	0.138	0.71	7.1%	24.1%	1.137	0.286	25.0%	21.0%	0	1	18.1%	50%	3.66	0.056
Streptomycin resistance rate	10.7%	11.1%	-	1	7.1%	11.4%	0	0.996	16.7%	9.9%	0.044	0.834	27.7%	40%	0.21	0.646
Ethambutol resistance rate	29.8%	22.2%	0.008	0.93	7.1%	32.9%	2.684	0.101	25.0%	29.6%	0	1	27.7%	40%	0.19	0.66
Kanamycin resistance rate	4.8%	0.0%	-	1	0.0%	15.1%	-	1	8.3%	3.7%	-	0.43	1.2%	30%	-	0.003
Capreomycin resistance rate	4.8%	11.1%	-	0.406	0.0%	6.3%	-	1	8.3%	4.9%	-	0.507	4.8%	10%	-	0.441
Para-aminosalicylic acid resistance rate	6.0%	22.2%	-	0.136	7.1%	7.6%	0	1	8.3%	7.4%	-	1	6.0%	20%	-	0.163
Protionamide resistance rate	7.1%	0.0%	-	1	0.0%	7.6%	-	0.586	16.7%	4.9%	-	0.171	3.6%	30%	-	0.015
Amikacin resistance rate	4.8%	0.0%	-	1	0.0%	5.1%	-	1	8.3%	3.7%	-	0.43	2.4%	20%	-	0.056
Ofloxacin resistance rate	21.4%	11.1%	0.087	0.768	14.3%	21.5%	0.067	0.796	25.0%	19.8%	0.001	0.97	16.9%	50%	4.16	0.041
Levofloxacin resistance rate	17.9%	11.1%	0.002	0.964	14.3%	17.7%	0	1	25.0%	16.0%	0.127	0.721	13.3%	50%	6.08	0.014
Moxifloxacin resistance rate	16.7%	0.0%	0.703	0.402	7.1%	16.5%	0.243	0.622	16.7%	14.8%	0	1	13.3%	30%	0.87	0.352
Prevalence of MDR-TB	19.0%	11.1%	0.017	0.895	7.1%	20.3%	0.631	0.427	16.7%	18.5%	0	1	15.7%	40%	2.1	0.148
Prevalence of PR-TB	11.9%	11.1%	-	0.944	7.1%	12.7%	-	1	16.7%	11.1%	-	0.63	10.8%	20%	-	0.336
Prevalence of MR-TB	13.1%	11.1%	0	1	7.1%	13.9%	0.07	0.791	0.0%	14.8%	0.936	0.333	14.5%	0%	-	0.35
Prevalence of DR-TB	44.0%	33.3%	-	0.727	21.4%	46.8%	3.132	0.077	33.3%	44.4%	0.526	0.468	41.0%	60%	-	0.318

MDR-TB, multidrug-resistant tuberculosis; PR-TB, poly-resistant tuberculosis; MR-TB, mono-resistant tuberculosis; DR-TB, drug-resistant tuberculosis.

## Data Availability

Data-sharing inquiries should be directed to the authors.

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
