# Peer review of "Trends in and Risk Factors for Drug Resistance in Mycobacterium tuberculosis in HIV-Infected Patients"

_viruses, 2024, doi:10.3390/v16040627_

Round 1
Reviewer 1 Report
Comments and Suggestions for Authors
Autors write an interesting and well present paper
Below my suggestions:
1. Introduction: add data on Global Tb report on HIV and TB co-infection and worsening condition. In addition, consider also as a risk factor of onset TB MDR the social determinant of health (see and cite Social determinants of therapy failure and multi drug resistance among people with tuberculosis: A review. Tuberculosis (Edinb). 2017 Mar;103:44-51. doi: 10.1016/j.tube.2017.01.002.)
2.Methods : clear and excellent
3. Result: no suggestion, clear table and figure are clear
4. Discussion: please discuss also the role of adverse event and drug drug interaction. Again the role of social determinat of health are relevant in HIV-TB coinfection. Add also limitations section
5. Conclusion: give some global health proposal that came from your good paper
Comments on the Quality of English Languagegood english minor mistake are present
Author Response
Response to Reviewers' Comments:
First of all, we are pleased to thank the reviewers for these valuable comments that help us improve the quality of our manuscript.
Reviewer 1
- Comments and Suggestions for Authors
Comment 1: Introduction: add data on Global Tb report on HIV and TB co-infection and worsening condition. In addition, consider also as a risk factor of onset TB MDR the social determinant of health (see and cite Social determinants of therapy failure and multi drug resistance among people with tuberculosis: A review. Tuberculosis (Edinb). 2017 Mar; 103:44-51. doi: 10.1016/j.tube.2017.01.002.)
Response: We are very grateful for pointing out the deficiencies of this article and offering improvement approach. In the section of INTRODUCTION, we refreshed the reference of Global tuberculosis report 2023 and added the latest epidemiological data of HIV/TB coinfection (lines 40-50 in the version without tracked changes). As our study did not include the social determinants of health as the risk factors, we consider that it is inappropriate to add related literature in this part of background introduction, but we have added it in the DISCUSSION section.
Comment 2: Methods: clear and excellent
Response: We thank the reviewer for the positive feedback.
Comment 3: Result: no suggestion, clear table and figure are clear
Response: Much thanks for your recognitions of our work.
Comment 4: Discussion: please discuss also the role of adverse event and drug drug interaction. Again the role of social determinat of health are relevant in HIV-TB coinfection. Add also limitations section
Response: Thanks for your constructive suggestions. In the DISCUSSION section in the revised manuscript, we discussed the role of adverse event and drug-drug interaction on impact of acquired drug resistance in lines 288-303 of the version without tracked changes and added the content of social determinant as the potential factors of MDR-TB in lines 306-311. Furthermore, we added the fourth point of the limitations in lines 350-353.
Comment 5: Conclusion: give some global health proposal that came from your good paper
Response: We quite appreciate for your insightful suggestion. In this section of revised paper, we elucidated the meaning of our study and provided several proposals on the management of drug resistance tuberculosis in lines 360-364.
- Comments on the Quality of English Language
good english minor mistake are present
Response: Thanks quite a lot for your helpful comment. Again, we have checked all the expressions in the manuscript and revised the language errors.
Reviewer 2 Report
Comments and Suggestions for Authors
Le et al. reported on mycobacterial resistance in Chinese HIV patients. The epidemiological report is short and straightforward, I only have very few minor suggestions.
1. Methods chapter, sub-heading 2.3.) Why did the authors decide against simply using WHO definitions? Please explain more thoroughly, as deviation from WHO standards makes the comparison of your work with other international publications unnecessarily challenging.
2. Discussion, limitations paragraph, line 296) I guess the term “can” is too tough wording here and should be replaced by “might”.
Author Response
Response to Reviewers' Comments:
First of all, we are pleased to thank the reviewers for these valuable comments that help us improve the quality of our manuscript.
Reviewer 2
Comments and Suggestions for Authors
Comment 1: Methods chapter, sub-heading 2.3.) Why did the authors decide against simply using WHO definitions? Please explain more thoroughly, as deviation from WHO standards makes the comparison of your work with other international publications unnecessarily challenging.
Response: We are extremely grateful for your constructive and enlightening suggestions. The biggest difference between the Chinese standards of tuberculosis classification (WS 196-2017) and the latest WHO standards is about the definition of XDR-TB. The reason why we adopted the former standards to define XDR-TB is mainly because phenotype susceptibility testing of linezolid and bedaquiline has not been carried out and the results were unavailable in our hospital. We have made further explanations in the manuscript correspondingly (lines 135-137 in the version without tracked changes). We renewed the references about WHO definitions and changed the sequence of literatures accordingly.
Comment 2: Discussion, limitations paragraph, line 296) I guess the term “can” is too tough wording here and should be replaced by “might”.
Response: Thanks a lot for this meaningful and wise advice. We have replaced the term “can” with “might” in the revised manuscript.